# Meridianins and Lignarenone B as Potential GSK3β Inhibitors and Inductors of Structural Neuronal Plasticity

**DOI:** 10.3390/biom10040639

**Published:** 2020-04-21

**Authors:** Laura Llorach-Pares, Ened Rodriguez-Urgelles, Alfons Nonell-Canals, Jordi Alberch, Conxita Avila, Melchor Sanchez-Martinez, Albert Giralt

**Affiliations:** 1Department of Evolutionary Biology, Ecology and Environmental Sciences, Faculty of Biology and Biodiversity Research Institute (IRBio), Universitat de Barcelona, 08028 Barcelona, Catalonia, (Spain); lllorapa7@alumnes.ub.edu (L.L.-P.); conxita.avila@ub.edu (C.A.); 2Mind the Byte S.L., 08007 Barcelona, Catalonia, Spain; agtagtagt@hotmail.com; 3Departament de Biomedicina, Facultat de Medicina i Ciències de la Salut, Institut de Neurociències, Universitat de Barcelona, 08036 Barcelona, Spain; enedrodriguez@ub.edu (E.R.-U.); alberch@ub.edu (J.A.); 4Institut d’Investigacions Biomèdiques August Pi i Sunyer (IDIBAPS), 08036 Barcelona, Spain; 5Centro de Investigación Biomédica en Red sobre Enfermedades Neurodegenerativas (CIBERNED), 28031 Madrid, Spain; 6Production and Validation Center of Advanced Therapies (Creatio), Faculty of Medicine and Health Science, University of Barcelona, 08036 Barcelona, Spain; 7Molomics, 08028 Barcelona, Spain

**Keywords:** Computer-aided drug design, Alzheimer’s disease, marine natural products, neural plasticity, *Aplidium* tunicates, *Scaphander* molluscs

## Abstract

Glycogen Synthase Kinase 3 (GSK3) is an essential protein, with a relevant role in many diseases such as diabetes, cancer and neurodegenerative disorders. Particularly, the isoform GSK3β is related to pathologies such as Alzheimer’s disease (AD). This enzyme constitutes a very interesting target for the discovery and/or design of new therapeutic agents against AD due to its relation to the hyperphosphorylation of the microtubule-associated protein tau (MAPT), and therefore, its contribution to neurofibrillary tangles (NFT) formation. An in silico target profiling study identified two marine molecular families, the indole alkaloids meridianins from the tunicate genus *Aplidium*, and lignarenones, the secondary metabolites of the shelled cephalaspidean mollusc *Scaphander lignarius*, as possible GSK3β inhibitors. The analysis of the surface of GSK3β, aimed to find possible binding regions, and the subsequent in silico binding studies revealed that both marine molecular families can act over the ATP and/or substrate binding regions. The predicted inhibitory potential of the molecules from these two chemical families was experimentally validated in vitro by showing a ~50% of increased Ser9 phosphorylation levels of the GSK3β protein. Furthermore, we determined that molecules from both molecular families potentiate structural neuronal plasticity in vitro. These results allow us to suggest that meridianins and lignarenone B could be used as possible therapeutic candidates for the treatment of GSK3β involved pathologies, such as AD.

## 1. Introduction

Glycogen Synthase Kinase 3 (GSK3) is a key contributor to the abnormal phosphorylation of the microtubule-binding protein tau in the process thought to cause neurofibrillary tangle (NFT) formation in Alzheimer’s disease (AD) [1,2]. GSK3 is an ubiquitous serine (Ser)/threonine (Thr) protein kinase and is involved in the transfer of a phosphate group from adenosine triphosphate (ATP) to Ser and Thraminoacid residues of target substrates. GSK3 is constitutively active, its substrates usually need to be pre-phosphorylated by another kinase, and it is inhibited, rather than activated, in response to stimulation of the insulin and Wnt pathways [3,4,5]. There are two highly conserved isoforms of GSK3, GSK3α and GSK3β. Particularly, GSK3β is widely present in the brain and is associated with several neurodegenerative diseases, including Parkinson’s disease (PD), AD and Huntington’s disease (HD) [6,7,8,9]. The predominant hypothesis in AD suggests that the activity of phosphatases and kinases, in particular GSK3β, is affected by amyloid peptides. Changes in kinase activity of GSK3β are an intrinsic aspect of the pathological problem in AD, as they negatively affect, even interrupting, synaptic signals essential for learning and memory [10]. GSK3 activity can be regulated by serine 9/21 phosphorylation. The kinase can be phosphorylated at additional different sites, but their regulatory outcomes remain unclear [3]. 

In AD, GSK3β is commonly regulated by inhibitory phosphorylation on Ser9, located at the N-terminal tail. The dysregulation of this process results in a GSK3β permanent abnormal activation that in turn induces a tau hyperphosphorylation leading to its aggregation [7,11,12,13]. From a drug development perspective, the potential therapeutic strategies aimed to target GSK3β are oriented to the reduction of tau hyperphosphorylation by its inhibition. Significant efforts have been made in the past years to design new and selective GSK3 inhibitors, acting over the ATP catalytic pocket or over other allosteric cavities [14]. However, most of the obtained compounds considered as hits or starting points have not advanced to the clinic because of administration, distribution, metabolism, excretion and toxicity (ADMET) problems [15]. In fact, some of the early GSK3β inhibitors that entered into clinical trials failed for toxicity problems or because off-target interactions [16,17]. Concretely, some of the main problems were: (1) Too high doses required to achieve brain penetrance causing in turn off-target effects in other tissues such as the musculoskeletal system or (2) to be unable to inhibit GSK3β in humans [18]. Such undesired and off-target effects would be due to the broad spectrum of GSK3β functions and the lack of selectivity on its kinase activity by these early compounds provoking exaggerated constitutive activity inhibition [16]. More recently, only a few potential inhibitors reached clinical trials in human subjects with AD or other diseases such as cancer. Unfortunately, compounds such as LY2090314 and Tideglusib showed no therapeutic effects [19,20,21]. Others such as Enzastaurin, induced unacceptable toxicity effects in patients with glioma or ovarian cancer [22,23]. Finally, lithium was among the most promising compounds to treat AD but inconclusive results have been found with some studies reporting no effects in AD patients [24] or even toxic effects in elderly AD patients [25]. Thus, there is still a clear need to develop better and safer GSK3β inhibitors.

Marine natural products, comprising a huge variety of chemical structures and being a serendipitous source of new molecules, could play a key role on this need [26,27,28,29,30,31]. In fact, the biomedical and pharmacological potential of marine natural products is known to be still underexplored [32,33]. In a previous study of our group, aimed to find possible molecular targets for a set of marine natural products, we observed that some of them can interact with proteins involved in neurodegenerative diseases. According to our interests, two of them were found particularly interesting as potential therapeutic agents against GSK3β: meridianin A and lignarenone B (Figure 1) [34,35]. See Figure 1 for a graphical summary of the study.

Meridianins are a family of indole alkaloids isolated from marine benthic organisms from Antarctica [36,37]. These ascidian’s natural products consist of an indole framework linked to an aminopyrimidine ring. Lignarenones, isolated from a gastropod mollusc from the temperate waters of the Mediterranean Sea, are two phenyl conjugated trienones, also classified as polyketides (Figure 2). These compounds are isolated from specimens of the tunicate genus *Aplidium* Savigny, 1816 and the shelled cephalaspidean mollusc *Scaphanderlignarius*(Linnaeus, 1758), respectively [36,37,38,39].

Supporting these findings, a large number of heterocyclic inhibitors of GSK3 have been identified in the past few years and can be classified as ATP-competitive and non-ATP-competitive [14,40,41,42]. Interestingly, the chemical structures of most of these marine inhibitors possess an heterocyclic scaffold, similar to the indole scaffold of meridianins (Figure 2). However, not all of them are heterocyclic compounds, for example, palinurin and tricantin, are sesquiterpenes which present a linear nature, together with aromatic cycles in the terminal part, similar to lignarenones (Figure 2). It is noteworthy that these compounds have been synthesized in the past and this is relevant for the potential development of new derivatives from both meridianins and lignarenones [43,44,45,46].

In this work, we analysed the putative inhibitory activity of meridianins and lignarenones (computational studies were performed only over lignarenone B). We aimed to ascertain whether these compounds were able to inhibit GSK3β activity. To do so, a computational approach using docking calculations and molecular dynamic (MD) simulations was performed with the aim of elucidating the ability of meridianins and lignarenones to act as ATP competitive or non-ATP-competitive inhibitors. In the first case, calculations were done over the ATP cavity and in the second case over the substrate pocket, finding after a possible allosteric binding cavities detection study. Next, an experimental validation of the inhibitory activity in vitro, using primary cortical neuron cultures, and its potential to induce structural neuronal plasticity of both molecular families was performed. 

## 2. Materials and Methods 

### 2.1. Computational Analysis 

#### 2.1.1. Target Selection and Modelling

From all the available structures of human GSK3β in the Protein Data Bank (PDB), the 3-dimensional (3D) crystallographic structure 6B8J with the co-crystalized ligand 65C, was selected, and thus the protein structure modelling from it [47,48]. The meridianins and lignarenone B structures were modelled from the 2D chemical structure previously published [36,37,38,39]. 

#### 2.1.2. Cavity Search

Fpocket software, a protein pocket prediction algorithm, was used to identify different cavities on the surface of GSK3β [49,50]. 

#### 2.1.3. Docking Calculations

Docking calculations were performed over the two top ranked pockets in the cavity search: the ATP cavity and the substrate pocket. The modelled protein structures and the marine molecules were employed as input of the calculations executed over each of the two cavities using Itzamna software tool [51]. More details on this approach can be found in the literature [29,30].

#### 2.1.4. Molecular Dynamics Simulation

Short MD simulations (1ns), using NAMD software version 2.11, were performed over the top ranked conformations obtained after docking, which were selected based on free binding energy to postprocess them [52]. Each ligand-target complex was protonated at physiological pH 7.4 and then placed into a transferable intermolecular potential 3P TIP3P water cubic box, imposing periodic boundary conditions, in which Na+ and Cl- ions were added to neutralize the charge of the system [53]. Electrostatic interactions were calculated by the particle-mesh Ewald method using constant pressure and temperature conditions. Each complex was solvated with a minimum distance of 10 Å from the surface of the complex to the edge of the simulation box. The temperature was maintained at 300 K using a Langevin thermostat, and the pressure was maintained at 1 atm using a Langevin Piston barostat. The time step employed was 2 fs. Bond lengths to hydrogens were constrained with the SHAKE algorithm [54]. Before production runs, the structure was energy minimized followed by a slow heating-up phase using harmonic position restraints on the heavy atoms of the protein. Subsequently, the system was energy minimized until volume equilibration, followed by the production run without restraints. The Amber ff99SB-ILDN and the General Amber Force Field (GAFF) sets of parameters were used to model the target and the marine molecules, respectively [55,56]. Ligand GAFF parameters were obtained using Antechamber, whereas the receptor structures were modelled using the leap module of Amber Tools [57,58]. 

#### 2.1.5. Molecular Dynamics Analysis

Visual inspection to each trajectory and the hydrogen bonds (HBs) occupancies analysis was performed using Visual Molecular Dynamics (VMD) software [59]. Thermodynamic (temperature, potential, kinetic and total energy) and structural (Radius of gyration (Rg), Root-mean-square deviation (RMSD) and Root-mean-square fluctuation (RMSF)) analysis were performed using GROMACS simulation package (see Appendix A) [60,61]. 

#### 2.1.6. Molecular Mechanics Generalized Born Surface Area

After performing MD simulations to estimate the binding free energy of GSK3β–marine molecules complexes, Molecular Mechanics Generalized Born Surface Area (MM/GBSA) reweighting techniques were employed [62,63,64]. MM/GBSA rescoring was performed using the MMPBSA python algorithm contained within the Amber Tools suit [65].

#### 2.1.7. ADMET Prediction

ADMETer, a software tool containing supper vector regression (SVR) and supper vector machine (SVM) ADMET machine-learning (ML) predictive models, was used to assess the ADMET properties (LogS, LogP, Caco2, blood-brain-barrier (BBB), plasma-protein binding (PPB), P-glycoprotein (Pgp), human ether-a-go-go gene hERG) of meridianin A-G and lignarenone B [29,66]. LogS reflects the solubility of the molecule in water (>5 indicate bad solubility). The P-glycoprotein (Pgp) predictor assesses whether a given compound is likely to be a substrate of Pgp. A high permeability is assessed in Caco2 when the predicted valued “high”. LogP allows us to estimate the distribution (>5 suggests good distribution) of a drug within the body (lipophilicity). Blood-brain-barrier (BBB) permeability describes the ability of a drug to cross into the brain. Plasma-protein binding (PPB) estimates the probability (>90% is considered high) that a given molecule binds to a plasma protein. The hERG predictors determine if a given compound is likely to be a hERG I/II inhibitor as the inhibition of potassium channels could result in fatal pathologies (values <4.0 estimates no toxicity). 

#### 2.1.8. Graphical Representations

Graphical representations were prepared using PyMOL version 1.7 and XMGRACE version 5.1.22 [67,68]. 2D images of marine molecules were prepared using RDKit python library [69]. 

### 2.2. In Vitro Analysis

#### 2.2.1. Marine Molecules

Marine compounds were obtained from the available sample collections at the University of Barcelona. The organisms were extracted with organic solvents and the extracts were further purified through chromatographic methods (HPLC) as previously reported [36,37,38,39]. Briefly, marine compounds were obtained from the available sample collections at the University of Barcelona. The organisms were extracted with organic solvents and the extracts were further purified through chromatographic methods (HPLC). Briefly, animals were extracted with acetone and sequentially partitioned into diethyl ether (repeated three times) and butanol fractions. Solvents were evaporated under reduced pressure, resulting in dry extracts used for chemical analysis. For meridianins, diethyl etherextracts were first screened by Thin Layer Chromatography (TLC), using Merck Kieselgel plates (20 × 10 cm and 0.25 mm thick), and light petroleum ether/diethyl ether (1:0, 8:2, 1:1, 2:8, 0:1) and chloroform/methanol (8:2) as eluents. The plates were developed with CeSO_4_. A yello wish UV–visible band at *R*f 0.63 (chloroform/methanol 8/2)corresponded with the fraction composed of the mixture of meridianins A–G. Extracts were further fractionated by molecular exclusion chromatography on Sephadex LH-20 and silica gel (Merck Kiesel gel 60, 0.063–0.200) columns by using chloroform/methanol 1:1 and a gradient of petroleum ether/diethyl ether as eluent, respectively. Fractions were further purified with HPLC techniques (Shimadzu with LC-10ADVP pump and SPD-10AVP UV detector) using reverse-phase semi preparative columns (Supelco Discovery^®^C18, 25 cm × 46 mm, 5 μm and 25 cm × 10 mm, Phenomenex, Kromasil C18) and water/acetonitrile and methanol/water 70:30 as solvent (flux 2 mL/min).For lignarenones, diethyl ether extracts were purified on normal-phase HPLC under isocratic elution with n-hexane/ethyl acetate (85:15) in 30 min (UV detection 256 nm). Meridianins were tested as mixtures in the experiments.

#### 2.2.2. Primary Cortical Neuron Cultures

Primary cultures of cortical neuronal were performed as previously described [70]. Cortex from E17.5 WT mouse embryos were dissected and gently dissociated with a fire-polished Pasteur pipette. Cells were seeded (50,000 cells/cm^2^ for immunochemical staining and 800,000cells/cm^2^ for Western blot analysis) onto 24 mm culture or 60 mm culture plates pre-coated with 0.1mg/mL poly-d-lysine (Sigma Chemical Co., St. Louis, MO, USA) and cultured in Neurobasal medium supplemented with B27 (Gibco, Paisley, Scotland, UK, 50×) and GlutaMAX (Gibco, 100×) at 37 °C in a humidified atmosphere containing 5% CO2. For biochemical assay, 60 mm culture dishes were treated at 7 days in vitro (DIV) with vehicle (PBS), meridianins (500 nM or 10 µM) and lignarenones (500 nM or 10 µM). Fifteen or 60 min after treatment, cells were washed with cold PBS and lysed for western blot (WB) analysis. For immunocytochemical staining and morphology analyses, 24 mm culture dishes were treated at 4 DIV with vehicle (PBS), meridianins (10 µM) and lignarenones (10 µM), and assessed 3 days later.

#### 2.2.3. Immunoblot Analysis

Cell samples were collected in cold lysis buffer containing 50 mM Tris base (pH 7.5), 10 mM EDTA, 1% Triton X-100 and supplemented with 1 mM sodium orthovanadate, 1 mM phenylmethylsulfonyl fluoride, 1 mg/mL leupeptin and 1 mg/mL aprotinin. Samples were centrifuged at 32,000× *g* for 15min and the supernatants collected. After incubation (1 h) in blocking buffer containing 10% non-fat powdered milk in Tris buffered saline-Tween (TBS-T) (50 mM Tris–HCl, 150 mM NaCl, pH 7.4, 0.05% Tween 20), membranes were blotted overnight at 4 °C with primary antibodies. Antibodies used (see Table 1) for immunoblot analysis were: GSK3β (1:1000; Cell Signalling, #9315), phospho GSK3β at Ser9 (1:1000; Cell Signalling, #9336xz) and α-Tubulin (1:40,000; Sigma-Aldrich, T9026). The membranes were then rinsed three times with TBS-T and incubated with horseradish peroxidase-conjugated secondary antibody for 1h at room temperature. After washing for 30 min with TBS-T, the membranes were developed using the enhanced chemilluminescence ECL kit (Santa Cruz Biotechnology). The Gel-Pro densitometry program (Gel-Pro Analyser for Windows, version 4.0.00.001) was used to quantify the different immunoreactive bands relative to the intensity of the α-tubulin or phospho GSK3β band in the same membranes within a linear range of detection for the ECL reagent.

#### 2.2.4. Immunocytochemical Staining

Immunochemical staining was performed following standard protocols available [71]. Briefly, primary cortical neuronal cultures were fixed at 7 DIV in 4% paraformaldehyde for 10 min. Cells were incubated overnight at 4 °C in 0.1 M PBS with 5% normal horse serum and with the proper primary antibody added (see Table 1) (MAP2; 1:1000, Sigma-Aldrich, M1406). After primary antibody incubation, cultures were washed with PBS and incubated 2h at room temperature with Alexa Fluor 488-conjugated AffiniPure donkey anti-mouse (1:100; Jackson Immunoresearch Laboratories, Inc., West Grove, PA, USA). Then, coverslips mounted with Fluoromount containing DAPI onto the surface of a slide after washes with PBS. Immunofluorescence images were taken using an Olympus B×60 epifluorescence microscope, using a 20× objective. 

#### 2.2.5. Imaging and Analysis

The in vitro Sholl analysis was performed with the freeware ImageJ (ImageJ, RRID:SCR 003070), we evaluated 45–55 neurons, all of them MAP2-positive from one primary cortical culture. 

### 2.3. Statistical Analysis

Statistical analysis was performed using one-way ANOVA with the Dunnett’s *post hoc* test as appropriate. Data analysis and graphs were created using Graphpad Prism Software version 6.0. A 95% confidence interval was used and values of *p <* 0.05 were considered as statistically significant. Data are expressed as mean ± S.E.M.

## 3. Results

### 3.1. Exploring Druggable Binding Sites on GSK3β

Using fpocket to analyse the GSK3β surface to identify novel allosteric binding sites, 15 plausible cavities were obtained (Figure 3A). Most of the pockets found here had been previously described over a different crystal structure [14], thus reinforcing the output of the performed cavity detection. Interestingly, looking at the obtained ranked pocket list (Figure 3A) and the corresponding images, the best cavity (number 1) is not the well-known ATP-binding site (number 2), but corresponds to the substrate binding site instead. Due to the interest in developing allosteric inhibitors and the high number of cavities found on the surface of GSK3β, it is relevant to study the capacity of meridianins and lignarenones to act as allosteric inhibitors as well as ATP-competitive inhibitors. From the fifteen cavities detected by fpocket, we focussed only on the ATP cavity and the substrate binding pocket (allosteric cavity) since, due to its druggability and structural properties, they become the two most suitable cavities to host a small molecule inhibitor. The location of the pockets and the amino acids involved in each cavity are shown in Figure 3B.

### 3.2. Binding of Meridianins and Lignarenone B to the ATP and Substrate Cavities

Docking calculations followed by MD simulations to post process them were performed in order to validate the proposed cavities and for evaluating the behaviour of the two marine molecular families as ATP or non-ATP-competitive inhibitors. One of the main characteristics of proteins is their flexibility, essential to carry out any function, but docking calculations do not used to consider this. Postprocessing the docking conformations by MDs is a good way to take this fact into account. Analysing the so called induced-fit events that allow the adaptation of the ligand to the target, and vice versa, (whereas docking, usually rigid, only allows the ligand movement), constitutes a well-established pipeline to study ligand-protein binding (docking+MD). The observed trend at docking results is confirmed after MD, being all the best energies corresponding to the molecules binding over the ATP cavity which are at least 6 kcal/mol higher than those obtained on the substrate pocket (although most are around 10 kcal/mol and on the particular case of meridianin F, reaching an 18 kcal/mol difference). The binding energies obtained after each calculation are summarized in Figure 4A.

Since MDs are dynamic processes, the number of hydrogen bonds (HBs) is not constant; they can be continuously forming and breaking, or they can be stable, depending on the system under study. In the case of GSK3β bound to meridianins A-G and lignarenone B, we found nine important HBs at the ATP cavity. These HBs are established with residues F67, V70, K85, D133, V135, R141, Q185, C199 and D200. Most of the listed aminoacidic residues are configuring the ATP binding pocket (Figure 4B) [16,29,30,72]. For the substrate binding pocket, we found sixteen important HBs. These were formed with F67, K85, K86, L88, Q89, F93, K94, N95, R96, R180, G202, S203, A204, K205, E211 and P212. All those residues are part of the substrate pocket (Figure 4C) [73]. In addition to the HBs, we also found that both, meridianins and lignarenone B, can interact with I62, F67, V70, A83, K85, E97, L132, T138, L188 and D200 via hydrophobic contacts on the ATP cavity and on the substrate pocket. Meridianins establish hydrophobic contacts with D90, F93, K94, N95, R96, K205 and I217, while lignarenone B does it with D90, F93, N95 and R96. In addition, two salt bridges were established on this pocket with residues D90 and E211 when meridianins are bound. 

In the ATP cavity, meridianins A-G and lignarenone B are placed on a very similar way, while on the substrate binding pocket the molecular placement of each compound varies with respect to the other, although with shared features (Figure 4B,C). One fact that could explain this pattern is the different dimension of the pockets, while the substrate pocket has a volume of 1808.60 Å3, the ATP pocket represents a quarter of its volume, 404.38 Å3, and thus, due to the size differences between meridianins and lignarenones, their position in a reduced space should be different.

RMSF analysis was performed to measure the amplitude of atom motions during the MD trajectories, elucidating the flexible regions of the proteins. We analysed the RMSF of GSK3β when meridianins and lignarenone B are bound over the ATP (Figure 5A) or substrate pockets (Figure 5B), obtaining homologous results. We found six peaks (corresponding to six different protein regions) of fluctuations with high mobility, with respect to the baseline. The first one is placed on residues 49-50, the second and the highest one between residues 91 to 94, the third involved the residue 124, the fourth involved residues 148 and 150, the fifth fluctuation occurs on residue 209 and the sixth is localized on residues 290 and 292. In the RMSF, when the marine molecules are bound to the substrate pocket, one more fluctuation on residue 66 can be observed.

### 3.3. Pharmacokinetic Properties Evaluation

It is well known that pharmacokinetics (PK) studies in the early stage of the drug discovery process play a key role in the development of new molecules, as they can predict the safety and efficacy of potential hit candidates, facilitating the appropriate lead compounds selection, saving investments in terms of money and time in expensive clinical trials [74,75]. Our experiments aimed to predict the absorption, distribution, metabolism, excretion and toxicity (ADMET) properties of potential therapeutic compounds, and thus, ADMET properties prediction for meridianins A–G and lignarenone B are reported (Table 2). Meridianins display some solubility problems (logS> 5 indicate not too much solubility), especially meridianin F, as well as difficulties to penetrate the BBB [29], a problem that is also shared by lignarenone B. This is also in agreement with the obtained logP values. Another issue is that all of the studied compounds present high PPB probability (Table 2). On the other hand, all of the compounds seem to be permeable, especially meridianins, according to Caco2 results (Table 2). Regarding Pgp binding, none of our compounds were predicted to act over it. Moreover, as another positive point, none of the compounds is an inhibitor of the potassium channels encoded by hERG.

### 3.4. Meridianins and Lignarenones Differentially Increased pGSK3β Ser9, but Not Total GSK3β Levels In Vitro

In order to confirm the predicted functional interaction between meridianins and lignarenone B with GSK3β, we used different doses of both marine molecules, at different times, to pharmacologically inhibit GSK3β. GSK-3β is regulated by post-translational phosphorylation of Ser9 (inhibitory)by multiple effectors such as p70^S6K^, p90^rsk^, protein kinase A (PKA), PKB (AKT), PKC isoforms and integrin-linked kinase (ILK) [76]. Phosphorylated Ser9 in the N-terminal domain of GSK-3β acts as a pseudo-substrate that blocks the access of substrates to the catalytic site. Therefore, we used the phosphorylation status of the GSK3β at Ser9 as a rapid, consistent and broadly used method to evaluate the activation state of the kinase [77,78]. Thus, primary cortical cultures were treated with vehicle, meridianins and lignarenones (500 nM or 10 μM) for two time points, 15 and 60 min (Figure 6). Western blot analysis was used to determine protein expression levels of total GSK3β and pGSK3β Ser9, its inhibitory phosphorylation site [79,80]. 

One-way ANOVA analysis indicated that meridianins treatment significantly increased pGSK3β levels both, after 15 min (F_2, 30_ = 4.189, *p =* 0.024) and after 60 min (F_2, 27_ = 6.892, *p =* 0.0038). Specifically, *post hoc* analysis revealed that at 60 min both doses, 500 nM (*p <* 0.05) and 10 μM (*p <* 0.01) significantly increased pGSK3β levels. Similarly, *post hoc* analysis after 15 min of treatment indicated that the dose of 10 μM (*p <* 0.01) but not the dose 500 nM exerted significant effects on pGSK3β levels (Figure 6A,B). Regarding to the treatment with lignarenones, one-way ANOVA analysis indicated that only after 15 min there was a significant change on pGSK3β levels (F_2, 34_ = 3.548, *p <* 0.05). *Post hoc* analysis indicated that this increase was only observed with the dose of 500 nM (*p <* 0.05). After 60 min of treatment with lignarenones, neither 500 nM nor 10 μM dose induced changes on pGSK3β levels (F_2, 28_ = 0.5814, *p* = 0.56) (Figure 6C,D). These results, far to be opposite, are complementary. On one hand, lignarenones have an acute effect within 15 min and then the effect decreases while, on the other hand, meridianins effects are more sustained over time. In the case of meridianins, the results are more robust in terms of inhibition. Regarding to the GSK3β total levels, they remain stable and no changes were detected.

### 3.5. Meridianins and Lignarenones Regulate Neurite Complexity in Vitro

To evaluate possible effects of meridianins and lignarenone B in neuronal structural plasticity, primary cortical neurons were treated at 4DIV with 10 μM of these marine molecules (highest dose) since it was the dose of meridianins with the best GSK3βinhibition capacity. As we previously described for this type of in vitro approach [70,81,82], three days after the treatment we analysed morphological characteristics of the imaged neurons stained for MAP2 by using the Sholl analysis (Figure 7). The results of meridianins, indicated that the number of intersections in the treated cultures were increased compared with the non-treated condition (two-way ANOVA analysis; interaction effect, number of dendrites, F_7, 536_ = 55.91, *p <* 0.0001) (Figure 7A,B). The results obtained for lignarenone B are similar to those obtained for meridianins and indicated that the number of intersections in the treated cultures were increased compared with the non-treated (two-way ANOVA analysis; interaction effect, F_7, 815_ = 7.247, *p <* 0.0001; group effect, F_7, 815_ = 67.90, *p <* 0.0001; number of dendrites, F_7, 815_ = 72.51, *p <* 0.0001) (Figure 7C,D). In both cases, positive results are obtained as there are increases of the neurite outgrowth. 

### 3.6. Effect of Meridianins and Lignarenones on Neuronal Viability

To elucidate potential secondary pharmacological effects induced by the treatments with meridianins or lignarenones, we next analysed cell viability on primary cortical cultures treated at 4DIV with the highest concentration (10 μM) of the drugs (Figure 8). Despite some previous works mentioning the toxicity of this sort of marine molecules at neuronal level, our results showed that 24 h after the treatment neither meridianins nor lignarenones induced changes in cell density in our primary cultures, meaning that cells were viable under the experimental conditions evaluated (one-way ANOVA analysis, F_2, 13_ = 1.600, *p* = 0.2392) (Figure 8A,B).

## 4. Discussion

GSK3β ATP catalytic pocket has been widely explored, often showing the specificity problems characteristic of protein kinases [83]. There are more than thirteen different GSK3β complexes available with a good resolution (lower than 2.5 Å) and after checking all of them, we selected the 6B8J crystal structure, representing human GSK3β together with CHIR99021, a selective inhibitor, released on 2017 [47]. This new PDB was not used in previous works where similar studies were performed [14]. Since crystal structures are biased to its co-crystals, perhaps using PDB new selective inhibitors and/or clues can be found. Allosteric cavities also open the possibility of designing inhibitors without these inconveniences. In that sense, the finding of the substrate pocket as a plausible drug-binding cavity is supported by literature data. For instance, manzamine A, a β-carboline alkaloid isolated from the marine sponges *Haliclona* and *Acanthostrongylophora*, or its derivatives, can bind to GSK3β near the pocket formed by the key residues R96, R180 and K205 (located at the predicted cavity 1) [84]. Moreover, some studies confirmed the interaction of manzamine A with the substrate pocket [72,85]. Our results are in agreement with the literature, reinforcing the election of pocket 1 as a good binding site for small molecules [86]. The docking post-processing by MDs allowed for the observation of induced fit events, as mentioned previously, and the existence or the magnitude of these events can be measured in different ways. The RMSF analysis revealed that when meridianin G and lignarenone B are bound to the substrate pocket, one more fluctuation on residue 66 can be observed with respect to the ATP cavity where this fluctuation is not observed in any of the analysed compounds binding. Another pattern detected is the fact that when lignarenone B is bound to any of both pockets, the fluctuation of GSK3β is in general higher than when meridianins bound, taking into account that induced fit can only involve minor conformational change in the overall protein [87], and this could be related to the structure of lignarenone B. Due to its size and shape, it could provoke larger changes in the protein binding site during MD to allow an optimal placement, improving the poses obtained during the docking process that could be more unstable than for smaller molecules. In addition to those findings, residue K94, present on the substrate pocket and a key component of the cavity, which establishes HBs with the marine molecules during the binding, is highly fluctuating. This can be explained because this residue is placed on the loop of the N-lobe, a very flexible region, that is very exposed to the solvent during the MD simulation [88]. Finally, substrate recognition requires GSK3β residues, F67, Q89 and N95, which facilitate the precise positioning of the substrate within the substrate binding pocket and provide an insight into the substrate binding and specificity [89]. To do so, the flexibility of the loop is necessary, and this is translated into high RMSF values of these residues and those next to them, as can be seen on peak 2 and the peak on the residue 66 during the binding of the meridianins and lignarenone B to the substrate pocket. It is worth to mention that although these results are very promising and are in agreement with the literature, the binding of meridianins to GSK3β has been proved as well as the binding of similar molecules to lignarenone B, further experimental validation is still needed to validate the in silico results [31,90,91].

PK studies pointed out that, as LogP values are lower than 5, the compounds have an appropriate hydrophobicity and permeability behaviour. However, to become drugs penetrating the central nervous system (CNS), molecules should have a LogP around 2 [92]. Some meridianins are almost there but most of our compounds are far of this optimal value. Another key distribution parameter, the PPB probability, also show high values, indicating that a high percentage of the administrated compound will be found attached to plasma proteins, affecting their diffusion and efficiency. Focusing on absorption parameters, the interaction with Pgp has many pharmacological implications that could result in pharmaceutical advantages or contraindications. For instance, Pgp modulation has been suggested as a mechanism to improve CNS pharmacotherapy, but it also plays a major role in multidrug resistance phenomenon (MDR)in cancer cells, depending on whether binding is as a substrate or as an inhibitor, and also on the isoform that the compound binds to [93,94,95,96]. Thus, none of our molecules seem to interact with Pgp, avoiding possible beneficial but also detrimental effects. Positive results were found on the toxicology prediction. hERG inhibition can lead to fatal pathologies such as cardiac diseases, and this is the principal cause of the development of acquiring long QT syndrome and fatal arrhythmia [97]. Not inhibiting hERG is a good and safe property of both meridianins and lignarenones.

In order to validate our in silico studies, we experimentally tested the two compounds, meridianins and lignarenone B in neuronal primary cultures. We focused on their capacity to inhibit GSK3β (directly or not) and to enhance structural neuronal plasticity since hyperactivation of GSK3β and neuronal atrophy are both core hallmarks in AD [7,98]. Our results showed a significant inhibition of GSK3β, being the meridianins-dependent inhibition more stable and consistent but being both of them capable to potentiate structural neuronal plasticity. Noteworthy, since we do not demonstrate a direct molecular link between both phenomena, we should consider the biochemical and morphological changes potentially independent. Several studies have shown that some GSK3β inhibitors can be toxic in chronic treatments [99]. However, these two marine drugs did not show any evident neurotoxic effects in vitro. All these parameters indicated that mostly meridianins but also lignarenone B accomplish with several of the main requirements to be potential therapeutic compounds for the treatment of AD [100]. Importantly, one of the main targets of GSK3β inhibition in the AD context is the correction of the Tau hyperphosphorylation [101], but we have not addressed this question in the present manuscript. Therefore, future studies should verify whether the compounds described here are also capable to prevent Tau hyperphosphorylation in the AD context. Overall, since challenging parameters related to its potential mechanism of action, druggability, pharmacokinetics, safety, capacity to inhibit GSK3β and induced structural neuronal plasticity have been overcome in the present work, future studies point out to test these compounds in preclinical mouse models of AD.

## 5. Conclusions

In this work, we have shown by computational and experimental studies that meridianins and lignarenone B are capable of inhibiting the activity of GSK3β, likely through an ATP competitive and non-competitive, allosteric mechanism. Furthermore, these marine molecules can increase neurite outgrowth in primary cortical neurons, without neurotoxicity. These compounds could be considered as hits, establishing a starting point to develop new future potential therapeutic agents for the treatment of AD. Future studies should be focused on the optimization of their absorption and distribution profiles, as well as their solubility, lipophilicity and BBB permeability. These compounds, once inside the cells, have shown a good inhibitory profile, and a good permeability toward the cellular membrane but they should be able to penetrate into the brain. Several strategies can be employed such as a proper modification of the chemical structure or its nano delivery, including also the possibility of become Pgp or other protein binders that facilitate their penetrance [102,103,104]. Altogether our results constitute a promising starting point for the development of novel anti-AD drugs.

## Figures and Tables

**Figure 1 biomolecules-10-00639-f001:**
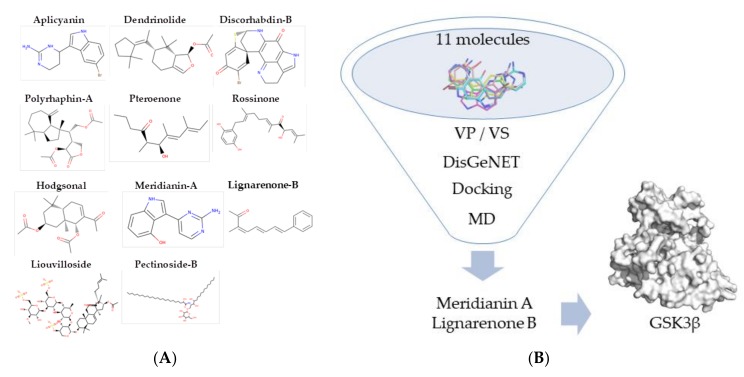
(**A**) List of previously isolated marine molecules. Structures of the eleven marine molecules selected for the initial study. (**B**) Workflow process. Graphical representation of the workflow process to disentangle Meridianin A and Lignarenone B as promising therapeutic molecules capable to inhibit Glycogen Synthase Kinase 3 (GSK3)β [34].

**Figure 2 biomolecules-10-00639-f002:**
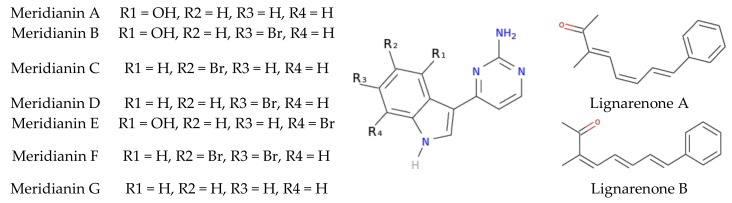
Structure of meridianins A-G and lignarenones A-B. Natural compounds isolated from marine organisms.

**Figure 3 biomolecules-10-00639-f003:**
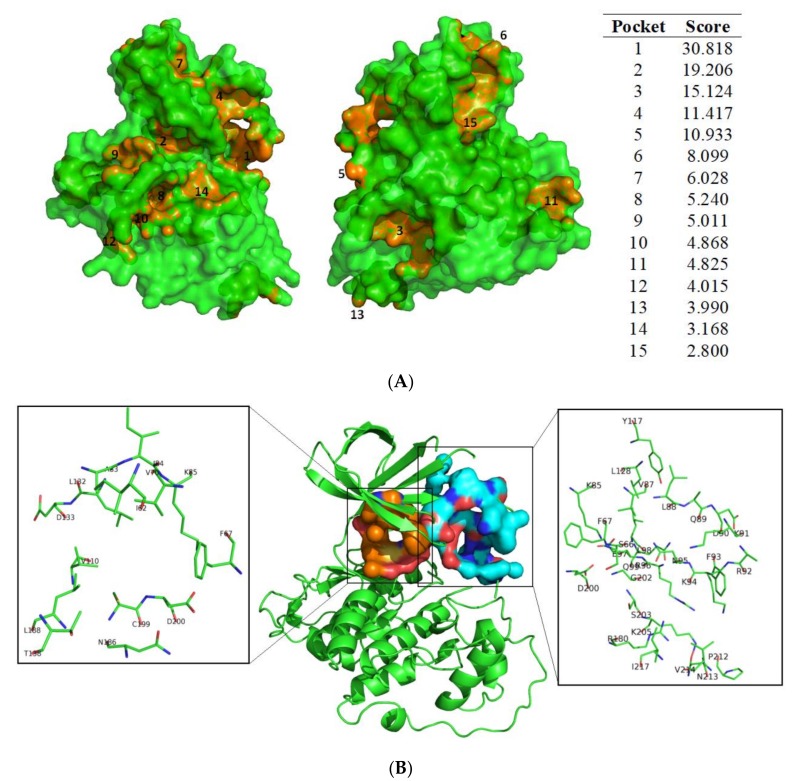
(**A**) Cavities found by fpocket and plot on surface. Numbers correspond to the ranking given by fpocket (1 is the best, and 15 the worst). In the list the score values of each pocket are shown. No worse details regarding the scoring function of fpocket can be found at Le Guilloux et al. (2009) [50]. (**B**) Pockets location and amino acids involved in each cavity. In the central image, a cartoon representation of the structure of GSK3β (PDB 6B8J) is depicted. In orange, the surface and the ATP-cavity are shown, while the blue surface represents the substrate binding pocket. In the left zoom image, all the amino acid residues that construct the ATP pocket are shown in sticks. In the right zoom image those residues that compose the substrate pocket are shown. Letters and numbers correspond to their position in the amino acid sequence and the 6B8J PDB file numbering.

**Figure 4 biomolecules-10-00639-f004:**
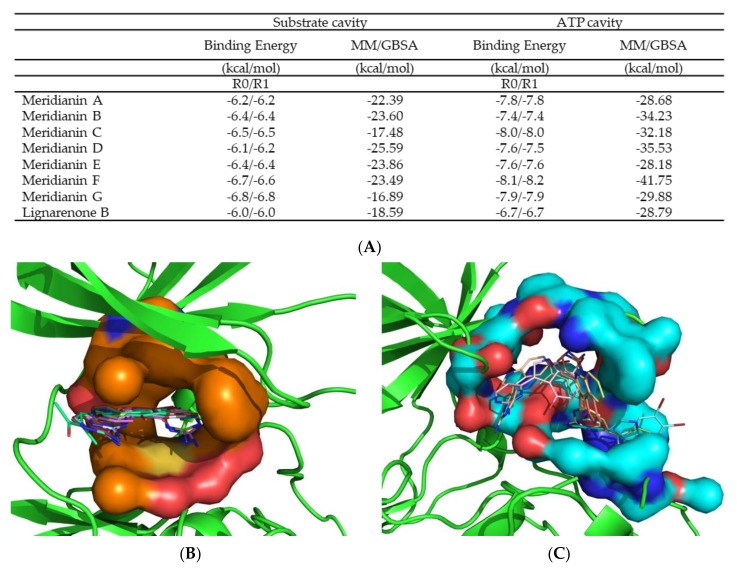
Representation of the two binding cavities ATP and substrate in surface, the binding mode of the marine molecules and the summary of the binding energy results. (**A**) Binding energy results after docking calculations and after 1 ns molecular dynamics (MD) simulations with molecular mechanics/generalised born surface area MM/GBSA calculations. All energies values are in kcal/mol. (**B**) The ATP pocket with all the meridianins and lignarenone B. (**C**) The substrate pocket also with all the meridianins and lignarenone B. Both images represent the last frame after MD simulation. Meridianin A-G colours: Peach, blue, tan, orange, pink, cyan and yellow. Lignarenone colour: green.

**Figure 5 biomolecules-10-00639-f005:**
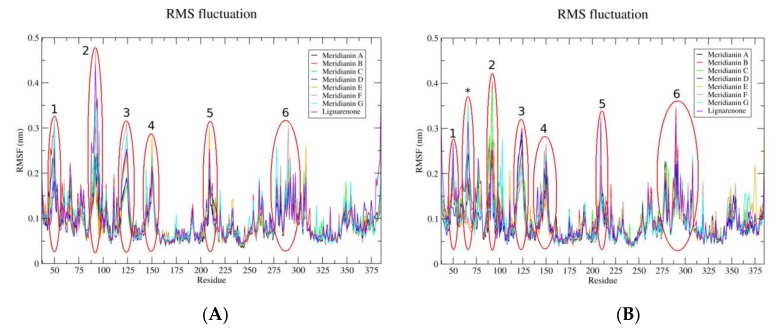
Root-mean-square fluctuation (RMSF) per residue (X-axis) values of each complex, GSK3β + marine molecules separated per pocket, along the molecular dynamic (MD) simulation. The highest fluctuations (>0.25 nm) detected are highlighted with red circles and those shared between RMSFs are numbered in appearing order. The asterisk represents a fluctuation >0.25 nm only observed on the substrate pocket. (**A**) On the left the RMSF of each system were marine molecules are bound to the ATP cavity and (**B**) on the right when the marine molecules are bound to the substrate pocket. The colour code for each system can be seen in the legend box.

**Figure 6 biomolecules-10-00639-f006:**
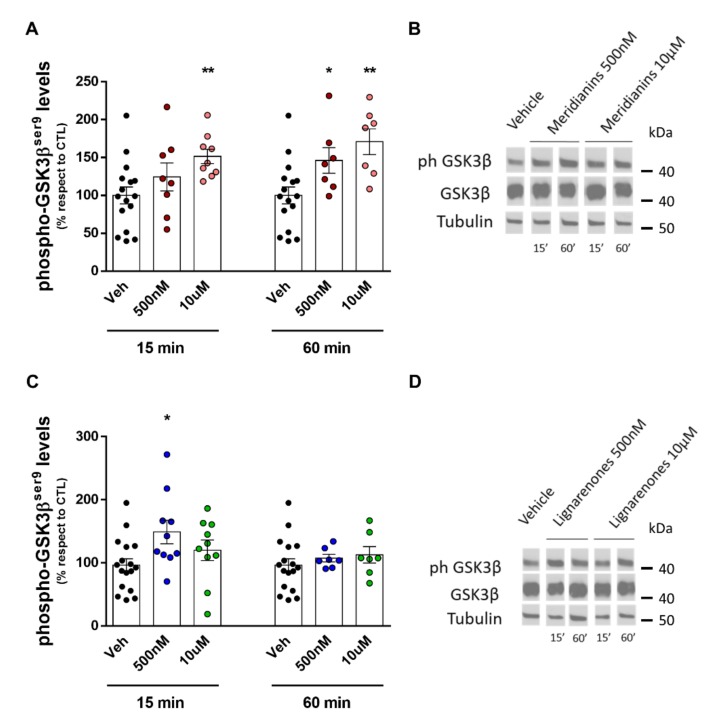
Increased GSK3β phosphorylation at Ser9 residue by meridianins and lignarenone B. Densitometric quantification of pGSK3β (**A**) and total GSK3β in cortical primary cultures treated with meridianins (500 nM or 10 μM) or vehicle, for 15 and 60 minutes. Protein levels were normalized with beta-tubulin as loading control. Data were expressed as (mean ± S.E.M). Data were analysed by one-way ANOVA followed by Dunnett’s test. * *p* < 0.05 and ** *p* < 0.01 compared with vehicle. (**B**) Representative immunoblots are shown. *n* = 16-9 cultures per condition at 15 min and 16-7 cultures per condition at 60 min. (**C**) Densitometric quantification of pGSK3β and total GSK3β in cortical primary cultures treated with lignarenone B (500 nM or 10 μM) or vehicle, for 15 and 60 min. Protein levels were normalized with beta-tubulin as loading control. Data were expressed as (mean ± S.E.M). Data were analysed by one-way ANOVA followed by Dunnett’s test. * *p* < 0.05 compared with vehicle. (**D**) Representative immunoblots are shown. *n* = 17–10 cultures per condition at 15 min and 17–7 cultures per condition at 60 min.

**Figure 7 biomolecules-10-00639-f007:**
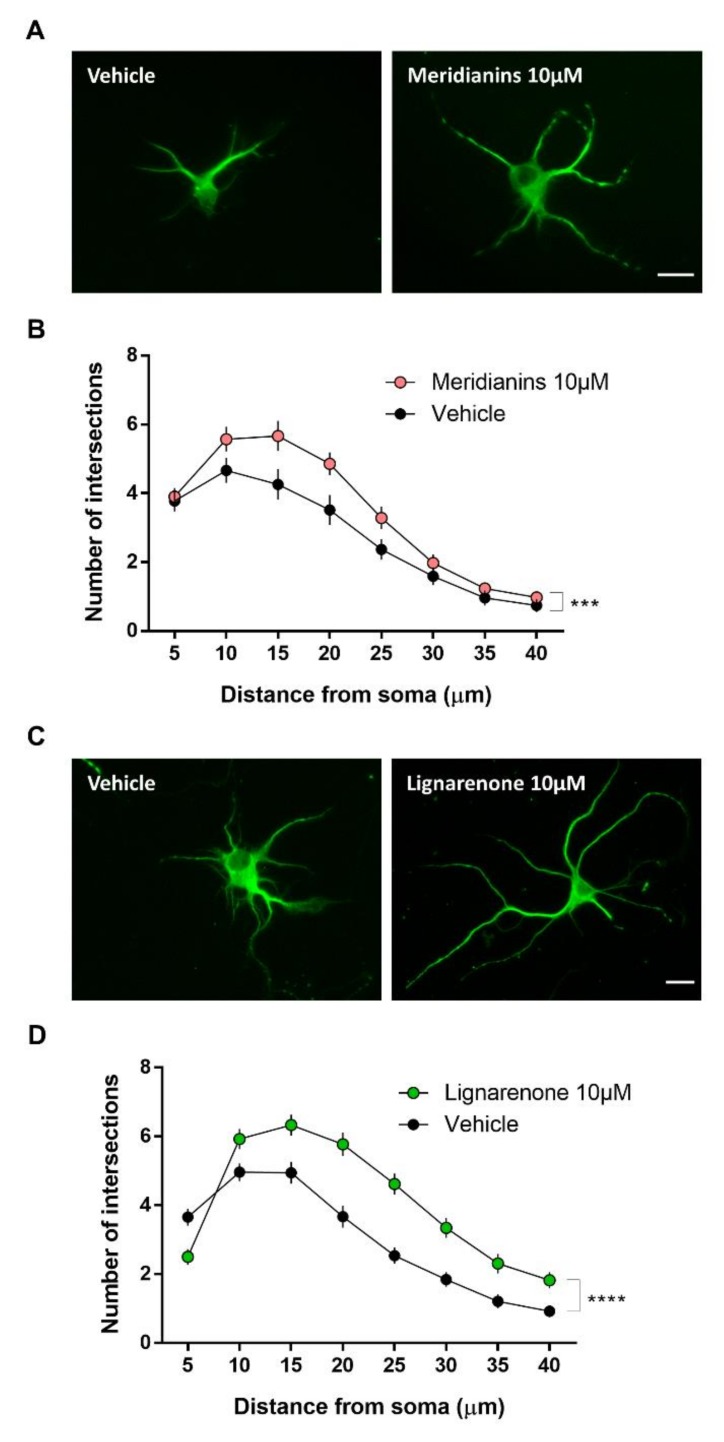
Meridianins and lignarenone B up-regulate neurite complexity in vitro. (**A**) Representative MAP2 images obtained by epifluorescent microscopy from primary cortical neurons treated with vehicle or meridianins 10 μM. Scale bar, 40μm. (**B**) Sholl analysis from MAP2-positive neurons (two-way ANOVA analysis; interaction effect, number of dendrites, F_7, 536_ = 55.91, *p <* 0.001). *n* = 45–55 neurons per condition. (**C**) Representative MAP2 images obtained by epifluorescent microscopy from primary cortical neurons treated with vehicle or lignarenone B 10μM. Scale bar, 40μm. (**D**) Sholl analysis from MAP2-positive neurons (two-way ANOVA analysis; interaction effect, F_7, 815_ = 7.247, *p <* 0.0001; group effect, F_7, 815_ = 67.90, *p <* 0.0001; number of dendrites, F_7, 815_ = 72.51, *p <* 0.0001). *n* = 45–5 neurons per condition. *** *p* < 0.001 and **** *p* < 0.0001 when compared with Vehicle.

**Figure 8 biomolecules-10-00639-f008:**
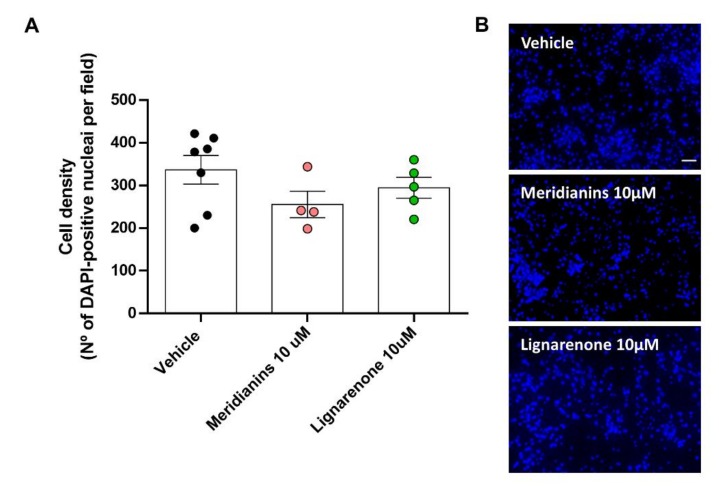
Cell viability is not affected by the highest dose of both, meridianins and lignarenone B. (**A**) Cell count by DAPI staining shows no significant differences between groups (One-way ANOVA analysis, F_2, 13_ = 1.600, *p* = 0.2392). (**B**) Representative DAPI images obtained by epifluorescent microscopy from primary cortical neurons. Scale bar, 50μm. Data were expressed as (mean ± S.E.M). *n* = 10 fields counted/6 coverslips per condition.

**Table 1 biomolecules-10-00639-t001:** List of essential chemicals, reagents and antibodies used in the present study.

Reagent/Chemical/Antibody	Manufacturer	Reference
Neurobasal medium	Gibco	21103049
B27	Gibco	17504-044
GlutaMAX	Gibco	35050-038
Tripanblue solution	Sigma	T8154
Chemilluminescence ECL kit	Santa Cruz Biotechnology	sc-2048
Normal donkey serum	Jackson Immunores. Labs.	017-000-121
4% paraformaldehyde	Pierce	28908
Anti-GSK3β	Cell Signalling	#9315
Anti-phosphoGSK3βat Ser9	Cell Signalling	#9336
Anti-MAP2	Sigma-Aldrich	M1406
Anti-α-Tubulin	Sigma-Aldrich	T9026
Horseradish peroxidase-conjugated secondary antibody	Promega	W4021
Alexa fluor 488-conjugatedanti-mouse	Jackson Immunores. Labs.	715-545-150
DAPI Fluoromont	Southern Biotech	010020

**Table 2 biomolecules-10-00639-t002:** Summary of absorption, distribution and toxicity properties of meridianins and lignarenone B. Pgp: P-Glycoprotein, BBB: blood-brain-barrier, PPB: plasma-protein binding, hERG: human ether-a-go-go gene.

		Absorption			Distribution			Toxicity
	Mol Weight	logS	Pgp	Caco2	logP	BBB	PPB	hERG
Meridianin A	226.2	−4.2	inactive	High	1.5	NO	High	<4.0
Meridianin B	305.1	−5.0	inactive	High	2.4	NO	High	<4.0
Meridianin C	289.1	−5.6	inactive	High	3.1	NO	High	<4.0
Meridianin D	289.1	−5.6	inactive	High	3.1	NO	High	<4.0
Meridianin E	305.1	−5.0	inactive	High	2.4	NO	High	<4.0
Meridianin F	368.0	−6.2	inactive	High	3.6	NO	High	<4.0
Meridianin G	210.2	−4.5	inactive	High	2.4	NO	High	<4.0
Lignarenone B	212.3	−3.2	inactive	Low	3.6	NO	High	<4.0

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
