# Peer review of "Meridianins and Lignarenone B as Potential GSK3β Inhibitors and Inductors of Structural Neuronal Plasticity"

_biomolecules, 2020, doi:10.3390/biom10040639_

Round 1

Reviewer 1 Report

The authors have described molecular modeling studies and neuronal viability studies of a series of marine natural products Meridianins and lignarenone B. Some changes to the article are suggested here.

1) Include more detailed information on the failure of the clinical trial of GSK3b inhibitors for AD. Many of these inhibitors have also encountered failures in cancer therapeutics clinical trials. An in-depth analysis of these failures is required along with a comparison of the properties of present set of marine natural products would make this work stand out as to why we need more GSK3b inhibitors to be discovered.

2) The target for the present compounds are clearly neurodegenerative disorders, especially tau phosphorylation. Studies on the effect of these compounds on tau phosphorylation at specific residues that are phosphorylated by GSK3b would have greatly enhanced the significance of this study. There are published methods for such tau phosphorylation studies using neuronal cell lines and residue specific tau phosphorylation antibodies are available. Such a study would make this article much stronger.

Author Response

-We are very grateful to the reviewer for her/his suggestions and requirements. We have addressed all her/his points together with the points from the other reviewers and consequently we really think that sections such as the methods section and discussion/conclusions section have substantially improved. Altogether improved the quality of the entire manuscript.

The authors have described molecular modeling studies and neuronal viability studies of a series of marine natural products Meridianins and lignarenone B. Some changes to the article are suggested here.

1) Include more detailed information on the failure of the clinical trial of GSK3b inhibitors for AD. Many of these inhibitors have also encountered failures in cancer therapeutics clinical trials. An in-depth analysis of these failures is required along with a comparison of the properties of present set of marine natural products would make this work stand out as to why we need more GSK3b inhibitors to be discovered.

​-In agreement with the reviewer, we think that tackling his/her point by including more detailed information about the failure of the clinical trials of GSK3 beta inhibitors for AD or even cancer would stand out the need to perform more work in the discovery of new potential GSK3 beta inhibitors, specially natural/marine molecules. We have included such new information in page 2, lines 66-76.

2) The target for the present compounds are clearly neurodegenerative disorders, especially tau phosphorylation. Studies on the effect of these compounds on tau phosphorylation at specific residues that are phosphorylated by GSK3b would have greatly enhanced the significance of this study. There are published methods for such tau phosphorylation studies using neuronal cell lines and residue specific tau phosphorylation antibodies are available. Such a study would make this article much stronger.

​-This is a very interesting point raised by the reviewer but, unfortunately, to tackle this question, we should acquire these cell lines and perform several experiments in the laboratory. Since the COVID19 crisis has closed our lab we cannot perform such experiments. We apologize for such inconvenience and we hope this is not preventing the present work to be publishable in Biomolecules. Alternatively, we propose to comment the reviewer’s point in the discussion section as a limitation of the study and that future works should address it. Therefore, such corresponding changes have been introduced in page 18, lines 519-523.

Reviewer 2 Report

This manuscript describes an evaluation of several natural products as inhibitors of GSK3 beta. First, a protocol was used to evaluate potential cavities on the surface of GSK3 beta using a new x-ray crystal structure as a guide. Then several molecules were chosen for evaluation by docking and MD simulations. It is not clear why a rationale is not provided for the selected molecules other than citing a previous paper that indicated their interest in such molecules.

Their evaluation identified two pockets that, according to modeling, could accommodate all the molecules tested.

Next compounds were evaluated for their ability to induce the phosphorylation of GSK3 beta on Ser9, an AKT phosphorylation site. No rationale was provided for this. Finally, the compounds were also tested at one concentration and after three days, for their ability to induce morphological changes of primary cortical neurons.

Ultimately, the authors do not present an experimental study showing that any of the compounds interact with GSK3 beta, and they provide no rationale for why a compound might induce a change in GSK3 beta phosphorylation. The studies also offer no data linking a change in GSK3 beta activity to the alteration in the morphology of neurons.

Author Response

-We are very grateful to the reviewer for her/his suggestions and requirements. We have addressed all her/his points together with the points from the other reviewers and consequently we really think that sections such as the methods section and discussion/conclusions section have substantially improved. Altogether improved the quality of the entire manuscript.

1.This manuscript describes an evaluation of several natural products as inhibitors of GSK3 beta. First, a protocol was used to evaluate potential cavities on the surface of GSK3 beta using a new x-ray crystal structure as a guide. Then several molecules were chosen for evaluation by docking and MD simulations. It is not clear why a rationale is not provided for the selected molecules other than citing a previous paper that indicated their interest in such molecules.

-The question made by the reviewer is important and we agree that a stronger rationale should be provided. To tackle this point we have added a new figure 1 were we show a first table with several of the compounds isolated by our team and the workflow used to select the two promising/potential compounds finally described in the present manuscript. Changes have been added as a new figure 1, in page 3, lines 83-86 and the corresponding new figure legend.

Their evaluation identified two pockets that, according to modeling, could accommodate all the molecules tested.

2.Next compounds were evaluated for their ability to induce the phosphorylation of GSK3 beta on Ser9, an AKT phosphorylation site. No rationale was provided for this.

-We agree with the reviewer that a stronger rationale to use the ser9 as a marker of GSK3B activity should be provided. We have added in the text (page 12, lines 378-383) the following new sentences and references:

“GSK-3β is regulated by post-translational phosphorylation of Ser9 (inhibitory) by multiple effectors such as p70S6K, p90rsk, protein kinase A (PKA), PKB (AKT), PKC isoforms and integrin-linked kinase (ILK) (76). Phosphorylated Ser9 in the N-terminal domain of GSK-3β acts as a pseudo-substrate that blocks the access of substrates to the catalytic site. Therefore, we used the phosphorylation status of the GSK3B at serine 9 as a rapid, consistent and broadly used method to evaluate the activation state of the kinase (77, 78).”

 4.Finally, the compounds were also tested at one concentration and after three days, for their ability to induce morphological changes of primary cortical neurons.

-We agree with the reviewer that the rationale to use these conditions are not rigorously provided. Changes in the text including new sentences and references have been added to indicate why we used these conditions in page 14, lines 415-419. Briefly, we used only one dose for the morphological studies because were the ones with higher rate of GSK3B inhibition and, presumably, they could be the best to induce structural plasticity changes. The neuronal morphology studies were performed 3 days after the compounds bath application to better see potential changes in neuronal dendritogenesis as previously shown by our group performing this analysis between 3-7 days after drugs bath application.

Concretely, we have modified the text as follows:

“To evaluate possible effects of meridianins and lignarenones in neuronal structural plasticity, primary cortical neurons were treated at 4 DIV with 10μM of these marine molecules (highest dose) since it was the dose of meridinanins with the best GSK3 beta inhibition capacity. As we previously described for this type of in vitro approach (77, 78), three days after the treatment we analyzed morphological characteristics of the imaged neurons stained for MAP2 by using the Sholl analysis”,

5.Ultimately,1) the authors do not present an experimental study showing that any of the compounds interact with GSK3 beta, and 2) they provide no rationale for why a compound might induce a change in GSK3 beta phosphorylation. 3) The studies also offer no data linking a change in GSK3 beta activity to the alteration in the morphology of neurons.

Regarding the first question: This is a very interesting point. The referee is right, and a further experimental validation would be required to fully assure our results. Unfortunately, due to COVID19 crisis our lab is closed, and we cannot perform this kind of experiments. Alternatively, we have stated in the new version of the manuscript that the binding of meridianins to GSK3B has been demonstrated in the literature as well as the binding of a couple of molecules similar to LignarenoneB. Changes in the text have been added in the page 17, lines 488-491

Regarding the second question. We did not want to relate the compound’s function directly to the GSK3 beta phosphorylation. We used the phosphorylation levels of the serine 9 as a marker of the GSK3 beta activity (inhibition). We stated/clarified this in the page 12, lines 382-383 and in the page 18, line 510.

Regarding the later point (3rd question), we agree with the reviewer and, indeed, in the previous version of the manuscript we already avoided to directly link the two described phenomena: GSK beta inhibition and neuronal structural plasticity induction. We just evaluated both processes as a main potential targetable hallmarks in Alzheimer disease: GSK3 beta inhibition and recovery of the deficits in (structural) neuronal plasticity. In order to further clarify this point, we have added explanations in this sense, page 18, lines 513-515.

Reviewer 3 Report

The manuscript is well constructed and show the inhibitory potential of Meridianins and lignarenone-B against GSK3β. However, there are following minor comments need to be corrected:

Page 1; Line 27: Need to highlight the type of analysis

Page 1; Line 29: Significant results should be written in brief with numerical values.

Page 3; Line 95: The list of materials and chemicals are missing. Should be written with their resources.

Page 3; Line 96: Should be Computational Analysis.

Page 4; Line 154: It should be In vitro analysis

Page 4; Line 158: The HPLC analysis should be describe in brief even adapted from past studies.

Page 4; Line 164: What was the initial passage of the cells?

Page 4; Line 168: Did the cells were treated with penicillin/streptomycin when incubated for 7 days?

Page 5; Line 206: Subtitle should be numbered.

Author Response

We are very grateful to the reviewer for her/his suggestions and requirements. We have addressed all her/his points together with the points from the other reviewers and consequently we really think that sections such as the methods section and discussion/conclusions section have substantially improved. Altogether improved the quality of the entire manuscript.

The manuscript is well constructed and show the inhibitory potential of Meridianins and lignarenone-B against GSK3β. However, there are following minor comments need to be corrected:

Page 1; Line 27: Need to highlight the type of analysis

-We have specified the type of analysis. Page 1, lines 27-29.

Page 1; Line 29: Significant results should be written in brief with numerical values.

-We have re-written the significant results in brief and with their corresponding numerical values now in page 1, lines 31-32.

Page 3; Line 95: The list of materials and chemicals are missing. Should be written with their resources.

-Although all the materials are explained in a narrative style with their exact references, we agree with the reviewer that a list of essential materials and chemicals would facilitate the understanding of the methods by the potential readers. A new Table 1 has been added in page 7.

Page 3; Line 96: Should be Computational Analysis.

-We have changed this erratum.Now in line 124, page 4.

Page 4; Line 154: It should be In vitro analysis

-We have changed this erratum- Now in line 191, page 6.

Page 4; Line 158: The HPLC analysis should be describe in brief even adapted from past studies.

-We have added an HPLC materials and methods paragraph to better explain its methodology. Page 6, lines 195-214.

Page 4; Line 164: What was the initial passage of the cells?

-If we have correctly understood the question, we have to say that primary neurons are never passaged because they would die. They were seeded at the same day in vitro 0, when neurons were isolated from neural tissue, and they remained quiescent and they grew up in size and morphology, but they did not proliferate. Therefore, passages are not required.

Page 4; Line 168: Did the cells were treated with penicillin/streptomycin when incubated for 7 days?

-No

Page 5; Line 206: Subtitle should be numbered.

-We have added the number 2.3 following the enumeration.Now in line 265, page 8.

Round 2

Reviewer 1 Report

It is understandable that experiments are difficult to perform during the COVID-19 shutdown. The manuscript can be accepted in the revised form.

Reviewer 3 Report

All the comments have been addressed and manuscript improved from the previous version.